

# Reassessing public opinion of captive cetacean attractions with a photo elicitation survey

Sophia N. Wassermann[1,2], Edward J. Hind-Ozan[2,3] and Julia Seaman[4]

[1] Ryan Institute, National University of Ireland, Galway, Galway, Ireland
[2] Center for Marine Resource Studies, The School for Field Studies, South Caicos, Turks and Caicos Islands
[3] Sustainable Places Research Institute, Cardiff University, Cardiff, Wales, United Kingdom
[4] Babson Survey Research Group, Babson College, Babson Park, MA, United States of America

Corresponding author
Sophia N. Wassermann,
s.wassermann1@nuigalway.ie,
so@sowasser.com

## ABSTRACT

**Background**. Captive cetacean attractions are growing in number globally, their operators citing entertainment, education, and conservation as benefits. Those for and against developing such attractions claim public support. Previous public opinion research, however, shows little consensus, partly due to the introduction of biases in study design that influence participants' responses. Those involved in, or concerned with, developing and licensing these attractions need to better understand what drives the lack of consensus to take socially-acceptable decisions.

**Methods**. We reviewed previous research on public opinion of cetacean captivity, noting possible sources of bias. Survey question wording can be a major source of introduced bias, so we used an open-ended photo elicitation approach. We showed tourists in the Turks Caicos Islands ($N = 292$) images of a marine mammal park (MMP) killer whale show and a swim-with-the-dolphins (SWTD) attraction and asked for their qualitative comments on the potential development of each. They also indicated how likely they would be to visit each on a Likert scale.

**Results**. Respondents were generally against visiting MMP killer whale shows, with 60.9% not likely to visit. SWTD attractions were more popular; 60.3% were likely to visit. For SWTD, USA residents were more likely to visit; older respondents and those staying in all-inclusive resorts were less likely. Those staying in all-inclusive resorts were also less likely to visit MMP killer whale shows. The great majority of qualitative comments centred on either entertainment value or animal welfare concerns. There were very few, if any, comments on the education or conservation value of these attractions.

**Discussion**. Our findings contradict several previous studies on public opinion of captive cetaceans that did not use photo elicitation. The support shown for MMP killer whale shows in this survey was well below that claimed by studies conducted on behalf of captive cetacean attraction operators. Opposition to SWTD was also noticeably lower than that found in surveys conducted with wild cetacean tourism participants. This difference can likely be attributed to the different survey populations and settings, but this variation is also very likely attributable to researcher-introduced bias. While photo selection can introduce bias, photo elicitation reduces reliance on pre-scripted questions and responses, and seems to effectively reduce other forms of bias. Allowing open-ended responses, where participants responded to an image, seems to have given a more representative understanding of what is at the forefront of the public's mind than

closed questioning. These conclusions, among others made in this study, suggest that development decisions for captive cetacean attractions are being made on imprecise data. Going forward, data collected via responder-led, open-ended, bias-minimising approaches should at least be considered when informing such decisions.

# INTRODUCTION

Since the 1960s, thousands of cetaceans have been held captive in a globally-increasing number of marine mammal parks (MMPs), aquariums, and captive swim-with-the-dolphins (SWTD) attractions (*Jiang, Lück & Parsons, 2007*). In 2018, these included 60 killer whales or orcas (*Orcinus orca*) (*Orca Home, 2016*) and near 2,000 dolphins in upward of 300 facilities (*Born Free Foundation, 2018*; *Change for Animals Foundation, 2018*). There were approximately 20 SWTD attractions in the United States (US), 25 in the Caribbean, and numerous others in China, Japan, and other Asian countries (*Frohoff, 2003*; *Rose, Parsons & Farinato, 2009*; *Born Free Foundation, 2018*). The existence of such attractions has become increasingly controversial, with researchers, tourism industry actors, non-governmental organisations (NGOs), and segments of the public expressing strong pro- and anti-captivity viewpoints. Further development of captive cetacean attractions will depend on how these viewpoints influence policy-makers.

Current research on the benefits and concerns of captive cetacean attractions is mixed. While some research around cetacean captivity has found benefits to human participants and cetaceans, there is research demonstrating that these benefits are falsely-perceived or short-lived (*Orams, 1997*; *Reeves et al., 2003*; *Williamson, 2008*; *Morisaka et al., 2010*; *Tizzi, Accorsi & Azzali, 2010*; *Parsons et al., 2013*). Close encounters with captive cetaceans have been documented as educational, increasing visitors' awareness of conservation issues and their likelihood to advocate for the protection of wild cetaceans (*Alliance of Marine Mammal Parks and Aquariums, 1999*; *Ballantyne et al., 2007*; *Shani & Pizam, 2009*; *Harley, Fellner & Stamper, 2010*). Yet some contest the conservation benefits, stating that the removal of animals from the wild for use in attractions puts local cetacean populations at risk (*Fisher & Reeves, 2005*; *Parsons et al., 2010*). The potential for MMP and SWTD human-cetacean encounters to inspire change is also questioned; studies show that visitors to captive cetacean facilities learn little about conservation (*Barney, Mintzes & Yen, 2005*; *Curtin & Wilkes, 2007*; *Jiang, Lück & Parsons, 2007*; *Rose, Parsons & Farinato, 2009*; *Rechberg, 2011*; *Dougherty, 2013*). Likewise, while some research has demonstrated that these attractions benefit humans, providing entertainment (*Shani & Pizam, 2009*) and improving physical and psychological health (*Webb & Drummond, 2001*; *Brensing & Linke, 2003*; *Antonioli & Reveley, 2005*), other research has shown that any benefits are mediated by discomfort at the captive state of the animals and visitors finding the human-animal encounters too staged (*Curtin, 2006*; *Curtin & Wilkes, 2007*; *Jiang, Lück & Parsons, 2007*).

Additionally, the therapeutic efficacy of human-dolphin encounters has been disputed (*Marino & Lilienfeld, 2007*; *Fiksdal, Houlihan & Barnes, 2012*; *Marino & Lilienfeld, 1998*). Research shows that risks to human participants, such as physical harm and disease contraction from dolphins, is possible during SWTD encounters (*Mazet, Hunt & Ziccardi, 2004*; *Friend, 2006*). Other studies found that any benefits from contact are superficial and transitory (*Frohoff & Packard, 1995*; *Marino & Lilienfeld, 2007*; *Williamson, 2008*; *Hunt et al., 2008*; *Fiksdal, Houlihan & Barnes, 2012*).

Research on whether the benefits of cetacean captivity are outweighed by animal welfare concerns is also in contention. Some research has concluded that animal behaviour can be normal and welfare high when provided with adequate enrichment; recent research has shown that dolphins positively anticipate interacting with their trainers (*Perelberg et al., 2010*; *Tizzi, Accorsi & Azzali, 2010*; *Clegg, Borger-Turner & Eskelinen, 2015*; *Clegg, Van Elk & Delfour, 2017*; *Serres & Delfour, 2017*; *Makecha & Highfill, 2018*; *Clegg et al., 2018*). Yet animals in captivity have also been found to have increased stress levels, poor diet, a higher chance of injury, and, in the case of killer whales, higher mortality rates in captivity (*Kyngdon, Minot & Stafford, 2003*; *Whale and Dolphin Conservation Society & The Humane Society of the United States, 2003*; *Ugaz et al., 2013*; *Lott & Williamson, 2017*; *Jett & Ventre, 2015*). In the wild, dolphins ordinarily avoid human contact (*Constantine, 2001*; *Constantine, Brunton & Dennis, 2004*) and research has suggested that positive response to interactions with humans may be due to habituation, or a response to ostracism from a dolphin social group, rather than a common and enjoyed behaviour (*Kyngdon, Minot & Stafford, 2003*).

There is also limited agreement on the ethics of obtaining cetaceans for the attractions in question. Out of the 60 killer whales in captivity (as of 6 February 2018), 27 were wild-born and 33 were captive-born (*Whale and Dolphin Conservation Society, 2018*). For dolphins, some of those in SWTD attractions were captured from the wild, with 70% of dolphins in marine parks in Europe born in captivity (*European Association for Aquatic Mammals, 2016*). Some aquaria and captive cetacean attractions advocate that wild capture is an important conservation tool for threatened species (*Bossart, 2016*), but this assertion has been questioned by researchers citing the large number of individuals necessary for genetic diversity, limited available space, and high costs of captive breeding and reintroduction (*Curry, Ralls & Bronwell Jr, 2013*). Possible threats to the sustainability of local populations as a result of capture have been noted in Cuba and the wider Caribbean (*Waerebeek et al., 2006*; *Würsig, 2017*). Wild capture of cetaceans in Canada ceased in 1992, but smaller cetaceans have continued to be imported (*Tasker, 2018*). Recently, however, cetacean captivity was banned in Vancouver (*Lindsay, 2018*) and legislation has been put forward to ban cetacean captivity across Canada (*Lake, 2018*). In the United States (US), some attractions have ended their captive breeding programmes under the increased scrutiny (*Hacket, 2016*; *Bossart, 2016*). However, wild-capture and breeding programmes do persist globally, e.g., a killer whale breeding facility was recently opened in China and Russia set a quota of 13 captured killer whales for 2018, destined for newly-constructed captive cetacean attractions in China (*Actman, 2017*; *Master, 2018*).

The pro- and anti- arguments for keeping cetaceans in captivity have escalated in the last decade, with high-profile public debate over the ethical and conservation

implications of the practice (*Jiang, Lück & Parsons, 2007*; *Shani & Pizam, 2008*; *Thomas, 2017*; *Rose et al., 2017*). Traditional and online/social media have questioned the continued existence of captive cetacean attractions (*Coldwell, 2014*; *Kuo & Savidge, 2014*; *Lerer, 2014*; *Zimmermann, 2014*). The documentaries *The Cove* (*Pesman, Stevens & Psihoyos, 2009*) and *Blackfish* (*Cowperthwaite & Oteyza, 2013*), which raised questions about cetacean conservation, captive cetacean welfare, and killer whale trainer safety, have increased public scrutiny of captive cetacean attractions (*Rechberg, 2011*; *Parsons, 2012*; *Pernetta, 2014*; *Parsons & Rose, 2018*). In response, captive cetacean attraction operators have rallied to rebut criticism and improve animal welfare practices (*Alliance of Marine Mammal Parks and Aquariums, 2013*; *SeaWorld, 2014*; *Lange, 2016*).

There is varying public opinion towards visiting captive cetacean attractions. Industry polls in 1992 and 2005 found, respectively, that 89% and 97% of the general public thought aquaria (including MMPs and SWTD attractions) were important educational venues (AMMPA, 1992; *Alliance of Marine Mammal Parks and Aquariums, 1999*). *Jiang, Lück & Parsons (2007)* also stated that the majority of visitors to an MMP knew of educational opportunities and felt better-educated. Additionally, the 1992 survey found that 37% of respondents believed captivity to be detrimental to animal life spans (*Alliance of Marine Mammal Parks and Aquariums, 1999*). A 2003 Canadian poll, reported originally by Zoocheck (see *Jiang, Lück & Parsons, 2007*), and a 2014 US one showed public opposition to cetacean captivity at 68% and 50%, respectively (*Jiang, Lück & Parsons, 2007*; *Edge Research, 2014*). Whale-watching eco-tourists in Belize identified 96% opposition to the capture of dolphins, 78% opposition to keeping them in closed tanks, and 67% opposition to keeping them in open-sea pens (*Patterson, 2010*). A study in Aruba identified that only 35% of tourists would be as comfortable seeing dolphins in captivity as in the wild (*Luksenburg & Parsons, 2014*). Of tourists surveyed in the Dominican Republic, 70% had no plans to visit a captive dolphin facility (*Draheim et al., 2010*). In a 2004 survey of Canadian residents, the most common reasons given for not visiting captive cetacean attractions were lack of interest, high admission costs, and animal welfare issues. Respondents who visited these attractions cited the performances and educational opportunities, rather than human-animal contact, as their motives. Half of the visitors were knowledgeable of the associated animal welfare issues, but few were aware of any conservation concerns (*Jiang, Lück & Parsons, 2007*).

With such variable snapshots of public opinion on cetacean captivity, further studies are needed to more clearly inform attraction developers, cetacean conservationists, animal welfare advocates, and policy-makers. It has been suggested that these studies need to particularly address the introduction of bias in public opinion research on cetacean captivity, as previous research has often been seen as expending little or ineffective effort on the issue (*Marino et al., 2010*). We used a photo elicitation approach to research the opinions of tourists in the Turks and Caicos Islands (TCI) toward developing and visiting captive cetacean attractions. We also sought to gain insight on the lack of consensus in previous research on public opinion of cetacean captivity. Through using a methodological approach known for reducing the introduction of some forms of bias, we aimed to contribute to an accurate and up-to-date baseline of public opinion on cetacean captivity.

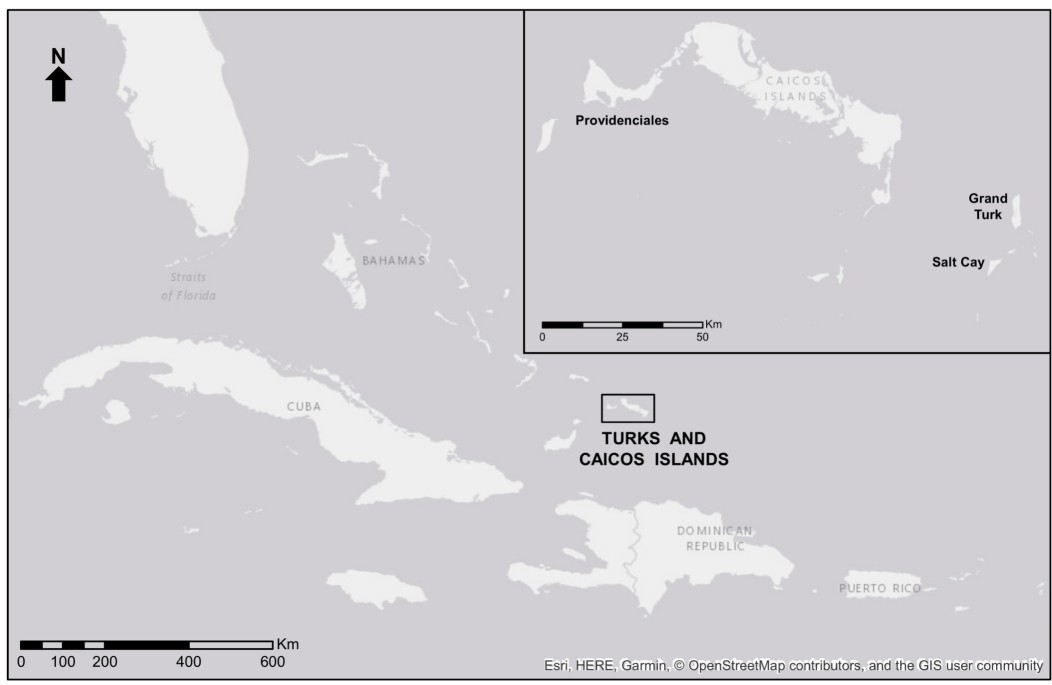

**Figure 1** **Map of the Turks and Caicos Islands.** Islands in the TCI associated with current or potential cetacean tourism. Map from Esri, HERE, GARMIN ©OpenStreetMap contributors, and the GIS user community. The data is available under the Open Database License, licensed as CC BY-SA.

## STUDY SITE

The TCI are an archipelago nation of approximately 40 islands (see Fig. 1) in the Caribbean region. With a growing population of 31,458 in 2012 (*Turks and Caicos Islands Government, 2012a*), the tourism sector was responsible for at least 41.8% of Gross Domestic Product (GDP) in 2011 (*Turks and Caicos Islands Government, 2012b*). Of the 1,315,268 tourists who visited the TCI in 2015, 70.7% visited the island of Grand Turk on cruise ships and most of the remaining 385,531 based their stopover[1] vacations on Providenciales (*Turks and Caicos Tourist Board, 2015*). Tourism has grown near year-on-year since at least the 1990s, a trend likely to continue (*Turks and Caicos Tourist Board & Department of Economic Planning and Statistics, 2009*; *Turks and Caicos Islands Government, 2012b*). The TCI Government (TCIG) encourages the development of attractions that will encourage further tourism (*Turks and Caicos Islands Government, 2012b*), but states that any industries supporting economic expansion should be "economically, culturally, socially and environmentally sustainable" (*Ministry of Finance Trade and Investment, 2013*).

Cetacean captivity was prohibited in the TCI until a 2012 legal amendment to the *Fisheries Protection Ordinance (1998)*, made to accommodate the development application for two proposed SWTD attractions (*Protests in TCI about possible dolphinaria, 2014*). This amendment was protested by environmental NGOs and the nation's Department of Environment and Marine Affairs (DEMA), highlighting conservation and animal welfare concerns (*Tyson, 2013*; *House of Commons Environmental Audit Committee, 2014*; *Protests*

[1] A 'stopover' tourist is defined as one who spends 24 h or more at their resort destination.

**Table 1  Types of bias potentially present in previous public opinion surveys relating to cetaceans.**

| Type of bias | Occurrence of bias | Sources |
|---|---|---|
| Sample | Where sample is from a population where for any reason that population is almost uniformly more informed than the general public on a public issue. Sample bias can exist when non-random samples are unintentionally enrolled as a result of respondent selection techniques. | *Berk (1983)*, *Marino et al. (2010)* |
| Motivated | Where researchers have a desired outcome, they can convey this to respondents through subtle communication during survey administration. Researchers can also insert their own bias by designing questions that they hope will either garner the responses they want, or that they will find interesting. Whilst insertion of this can be conscious, and perhaps as a result unethical, it can also be unconsciously inserted by well-meaning researchers. | *Hammersley & Gomm (1996)*, *Marino et al. (2010)* |
| Ingratiation | Respondents can adjust their answers to gain favour or avoid disagreement with researchers. They may adjust their answers to fit a hypothesis they believe the researcher to be investigating. The nature of questions and the manner or appearance of researchers can invite this kind of bias. | *Back & Gergen (1943) Dijkstra (1987)*, *Marino et al. (2010)* |
| Social desirability | Respondents may give answers that they believe to be socially desirable so that they appear to conform to a societal position they believe is seen as favourable. | *Rossiter (2009)* |

*in TCI about possible dolphinaria, 2014*). However, TCI policy-makers continued to back development, based on support from TCI citizens who hoped the facility would bring employment and on developer guarantees that the attractions would be especially popular with tourists from the US (*Dolphin Cove development to begin next year, 2014*; 'More jobs', 2014; *Protests in TCI about possible dolphinaria, 2014*; Tyson, 2014). Most cruise ship passengers and 81.7% of stopover guests in 2015 were US citizens (*Turks and Caicos Tourist Board, 2015*).

At the time of data-collection, the TCI SWTD attractions remained proposed but not constructed. The only existing tourism associated with cetaceans was small-scale whale-watching tours from Salt Cay. These tours did not ordinarily involve cruise ship tourists or Providenciales stopover guests (*Visit Turks & Caicos Islands, 2017*).

## MATERIALS & METHODS

Several types of bias can be present in public opinion surveys (see Table 1). Many of the previous studies of public opinion of cetacean attractions were conducted by researchers with their own opinions of cetacean captivity. While personal interest is a valid reason to conduct research (*Bennet, Ekinsmyth & Shurmer-Smith, 2002*), certain methods are inherently prone to researcher-introduced bias, even when the researcher is careful to avoid it. It was important for us, who ourselves identified as anti-captivity, to design a study that was as free of researcher bias as possible.

To avoid introducing potential conditioned bias developed through exposure to the intense local debate over the conservation, animal welfare, and job-creation issues

surrounding the development of the two local SWTD attractions, foreign tourists, rather than TCI residents, were chosen as respondents. In addition, the opinions of tourists are perhaps the most important when considering the justification for developing an SWTD attraction, as they will provide the attendance (or otherwise) that makes it viable. Motivated, ingratiation, and social desirability biases were minimised by designing a survey instrument that initially concealed the primary focus of the research from the respondent. Open-ended response options were favoured to minimise the chance introduction of various researcher biases during survey design. Open-ended responses preclude the collection of inaccurate data when respondents are forced to choose one closed option when they would rather choose multiple (*Zaller & Feldman, 1992*).

Our survey team first showed respondents a grid of six photographs (Fig. S1). These depicted six tourist attractions not present in the TCI, but that were popular elsewhere in the Caribbean region, according to feedback on the review website *TripAdvisor*. When shown the photographs, respondents were asked: "What are your opinions on any of these six attractions being introduced in the Turks and Caicos Islands?" This simple question did not introduce our own opinions or prior knowledge, and therefore personal bias, on any of the conservation, educational, entertainment, welfare or other issues associated with the attractions. No closed options were provided and respondents were not forced to comment on each photograph. Showing the six photographs simultaneously reduced bias associated with presentation order (*Gibson et al., 2014*). Surveyors took notes on the qualitative comments volunteered by respondents.

Photo elicitation has an excellent track record for accessing the true worldview of respondents (*Harper, 2002*), as it hands the role of dialogue construction, or the "voice of the research", to the research participant (*Frith et al., 2005*). Rather than taking verbal cues from the language used in researcher-designed questions, participants can reflect on what an image means to them in their own words. They may pick up on entirely different themes in a photograph than those that a researcher might expect (*Epstein et al., 2006*). Yet photo elicitation can still introduce bias, potentially motivated, when researchers do not theoretically account for variables between photographs (*Gaber & Gaber, 2004*). We justified selecting this method because it removed many further opportunities for insertion of researcher-induced bias and we took a theoretical approach to photograph selection to minimise introduction of our personal biases. Elimination of all variables in the images (e.g., the prominence of human subjects across all images) would not have accurately represented these attractions, nor likely have been possible. Therefore, we chose photographs from our personal collections and *Creative Commons* sources that best represented the perspectives, scenes, and human behaviour evident in a standard *Google Images* search for each attraction. Previous studies using vision-based approaches to selecting representative images have also used criteria such as the frequency of photographs depicting a scene and of the subject attention and behaviour represented (*Kennedy et al., 2007*). For example, the top 100 results returned for "swim-with-the-dolphins" included 81 close-ups of individuals swimming with captive dolphins, with 73 of those individuals facing the camera and clearly smiling. To further ensure internal validity, the attractions

were also named orally by our survey team when shown to the respondents, ensuring that focus was more likely to remain on the attraction.

After showing respondents the image grid, our surveyors asked, on a 4-point Likert scale, whether they would be "very unlikely", "unlikely", "likely", or "very likely" to visit such an attraction in the TCI. If we had asked these intention-orientated questions first, it is more likely that respondents would have only volunteered opinions to our open question that were aligned with their chosen closed response. Inclusion of a neutral option between "unlikely" and "likely" was considered but rejected, as we wanted to avoid potential social desirability bias causing respondents to choose uncontroversial options (*Garland, 1991*). The survey team were trained to ask questions in a neutral manner. They were also asked to avoid wearing clothing that might introduce ingratiation bias (e.g., t-shirts with anti-captivity or pro-conservation logos). Finally, respondents were asked demographic questions about their age, gender, country of residence, accommodation, and experience with and interest in cruise tourism.

We used a consecutive sampling approach (see *Lunsford & Lunsford, 1995*) to complete 292 daytime surveys with stopover tourists on Grace Bay Beach, Providenciales on 18 March 2014. With 292 respondents, we estimate the overall margin of error (95% confidence interval) to be ±5.7%, and results with confidence bounds that did not cross 50% were concluded to be "likely" or "unlikely" depending on the extent of their boundaries. As we had a large survey team, we could approach every visible tourist on the beach, with the exception of those engaged in activities that impeded their participation (e.g., swimming, sleeping). Respondents were told before they took the survey that their participation was optional and all who consented were enrolled. During the day, the great majority of Providenciales tourists are found on Grace Bay beach or in their resorts. As public access to resorts is generally prohibited, this was as complete and representative a sample of the target population as could be realistically achieved. This approach reduced the sample bias found in previous surveys of public opinion on cetacean captivity, where less strict formats of convenience sampling have been employed (*Marino et al., 2010*). Surveying in Grand Turk was not logistically possible, but we asked Providenciales visitors about their preference for cruise tourism to gauge the possible attitudes of the cruise ship passengers who visit Grand Turk.

Statistical analyses were conducted using R (*R Core Team, 2015*) and Prism. As the Likert scale used did not assign numerical values, we used non-parametric Chi-Square tests to assess hypotheses of difference. For testing the summary responses for each attraction, we used all four Likert variables. For testing on demographic variables, we condensed the responses to two groups, "likely" ("very likely" and "likely" responses) and "unlikely" ("very unlikely" and "unlikely" responses), to facilitate significance testing. We performed tests on a variable if there were large enough groups of individuals for detecting significance, defined here as greater than five individuals. We used Bonferroni corrections within the demographic subgroups (2–7 categories) and significance level is reported for the corrected *p*-value.

We used structural coding, as described by *Saldaña (2013)*, to code the surveyors' notes of respondents' qualitative responses. These were coded as either "NEGATIVE

OPINIONS OF MMPs" and "NEGATIVE OPINIONS OF SWTD ATTRACTIONS", or "POSITIVE OPINIONS OF MARINE MAMMAL PARKS" and "POSITIVE OPINIONS OF SWTD ATTRACTIONS". No neutral responses were expressed by respondents, so we did not include a neutral coding descriptor. We also conducted subcoding of the reasons for opinions where possible. We recorded the number of respondents expressing each opinion.

We followed all legal and ethical guidelines for conducting research in the TCI. We did not ask for personal identifiers during surveys, nor were they recorded if given. No individuals from vulnerable populations were enrolled. Verbal consent was acquired. Although the focus of research was initially concealed during survey administration to avoid introducing motivated, ingratiation, and social desirability biases, the true focus of the research (i.e., to measure public opinion of captive cetacean attractions) was revealed to respondents following their participation. No respondents subsequently withdrew their participation when the option was again offered.

## RESULTS

### Sample demographics

There were a total of 292 respondents and all responses were voluntary. Respondents were 61.1% female and 38.9% male ($n = 280$). By age, 15.2% of respondents were 18–29, with 10.5% being 30–39, 31.0% being 40–49, 23.1% being 50–59, and 22.2% being 60 or older ($n = 277$). Most respondents resided in North America, with 71.5% living in the US and 25.8% in Canada. The remaining 2.8% were from Europe, South America, and Egypt ($n = 291$). Where $n < 292$, it is due to non-responses, all of which are reported in Table 2.

In terms of tourists' preferences, those who would consider a future cruise vacation comprised 37.8%, with the remainder uninterested ($n = 288$). Of the tourists surveyed, 39.8% were staying in all-inclusive resorts, which provide activity programmes as part of the package, with the remainder staying in other accommodation ($n = 289$). Across the sample, 47.4% had vacationed in the TCI more than once ($n = 289$).

### Qualitative responses: rate and nature

Very few respondents offered qualitative responses for all six photographs. For the photograph of the SWTD attraction, 26.4% of respondents provided open-ended responses, with this reduced to 18.2% for the MMP killer whale show. While a small number of respondents responded in greater detail, most answers were between one and three sentences long. All qualitative responses were about the featured attractions, rather than comments that could only be attributed to the images themselves; no respondent remarked on the child in the SWTD image (Fig. S1).

### Overall perceptions of tourists

Respondents favoured the possibility of visiting a potential SWTD attraction over an MMP killer whale show, with an overall median description of "likely" to visit the SWTD attraction compared to "unlikely" for the MMP killer whale show. There was a significant difference ($p < 0.001$) between the responses for the MMP killer

**Table 2  The demographic composition of the 292 respondents.**

|  | Sub-Category | Count | Percent (%) |
|---|---|---|---|
| **Age (yrs)** | 18–29 | 42 | 11.7 |
|  | 30–39 | 29 | 8.1 |
|  | 40–49 | 86 | 24.0 |
|  | 50–59 | 64 | 17.8 |
|  | 60–69 | 40 | 11.1 |
|  | 70+ | 17 | 4.7 |
|  | No response | 14 | 3.9 |
| **Gender** | Male | 109 | 37.3 |
|  | Female | 171 | 58.6 |
|  | No response | 12 | 4.1 |
| **Residency** | USA | 208 | 71.2 |
|  | Canada | 75 | 25.7 |
|  | Other & no response | 9 | 3.1 |
| **Parental status** | Has children | 136 | 46.6 |
|  | Has no children | 155 | 53.1 |
|  | No response | 1 | 0.3 |
| **Visits to TCI** | Multiple | 152 | 52.1 |
|  | One | 137 | 46.9 |
|  | No response | 3 | 1.0 |
| **Interest in cruise tourism** | Have cruised/ Would again | 76 | 35.2 |
|  | Have cruised/ Would not again | 64 | 29.6 |
|  | Have never cruised/ Would cruise | 33 | 15.3 |
|  | Have never cruised/ Would not cruise | 115 | 53.2 |
|  | No response | 4 | 1.9 |
| **Accommodation type** | All-Inclusive | 115 | 39.4 |
|  | Other | 175 | 59.9 |
|  | No response | 2 | 0.7 |

whale show and the SWTD attraction. The preferred favourability rankings are SWTD attraction, aquarium, botanical gardens, craft market, MMP killer whale show, and maritime museum. The SWTD attraction, aquarium, and botanical gardens all had median descriptors of "likely" to be visited, with the MMP killer whale show and maritime museum having median descriptors of "unlikely". The craft market fell into its own significant group between "likely" and "unlikely". There was no significant difference between the SWTD attraction and the aquarium or botanical gardens, but there was a significantly higher likelihood to visit an SWTD attraction than a craft market ($p < 0.01$). The MMP killer whale show was significantly less attractive than the three "likely" attractions ($p < 0.001$) and the maritime museum ($p < 0.05$), but

**Table 3  Visitation likelihoods of TCI tourists to each attraction.**

| Attraction | Visitation Likelihood (%) | | | |
|---|---|---|---|---|
| | Very likely | Likely | Unlikely | Very unlikely |
| SWTD | 36.6 | 23.7 | 18.1 | 21.6 |
| MMP | 15.4 | 23.8 | 28.0 | 32.9 |
| Aquarium | 22.0 | 35.5 | 21.6 | 20.9 |
| Botanical Gardens | 22.2 | 34.4 | 23.3 | 20.1 |
| Maritime Museum | 5.9 | 23.7 | 30.0 | 40.4 |
| Craft Market | 17.4 | 31.4 | 26.1 | 25.1 |

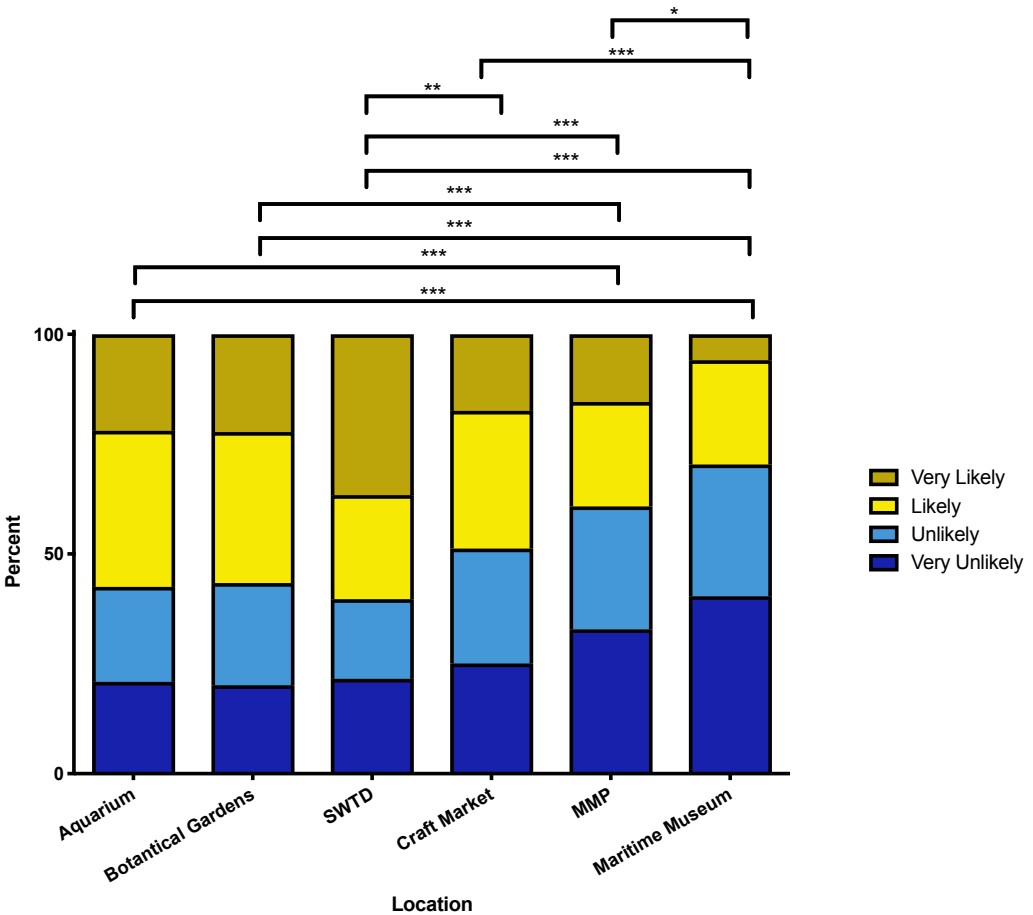

**Figure 2  Tourists' visitation likelihoods for the attractions.** Significant groupings of tourists' visitation likelihoods for the six attractions including swim-with-the-dolphins (SWTD) and marine mammal park (MMP). Asterisks summarise the value of $P$ more generally (*$P \leq 0.05$, **$P \leq 0.01$, ***$P \leq 0.001$).

not significantly less than the craft market. The full range of Likert responses for each attraction are reported in Table 3, with significant groupings shown in Fig. 2.

Only five respondents who were "likely" or "very likely" to visit an MMP killer whale show gave qualitative feedback, all stating "entertainment" as their reason for wanting

to visit. The 4.2% of those "likely" or "very likely" to visit and who provided qualitative feedback said that though they were disinclined to visit themselves, they would be likely to visit an MMP killer whale show because it would be entertaining for their children. For those "unlikely" or "very unlikely" to visit, and who offered qualitative comments ($n = 48$), the most frequently-given reasons were animal welfare concerns (72.9%), perceived over-commercialisation of the attraction (14.6%), and lack of entertainment value (10.4%). Their qualitative justifications for their decision-making included the belief that animals were being "abused" in such parks, that "animals [did not] belong in an environment like this", that they did not like the nature of performances, and that they objected to animals being "caged up". Respondents noted that there "was a lot of bad press" about killer whale shows and that *Blackfish* was "really sad". The documentary was cited by 14.6% of those "unlikely" or "very unlikely" to visit an MMP attraction and who gave qualitative responses for their reasoning for non-visitation. One respondent noted that their young daughter had told them the documentary showed abuse of killer whales. Only 4.2% of the same "unlikely" group, who provided qualitative responses, explicitly mentioned the human welfare threat to animal trainers as a reason for non-visitation.

For tourists offering qualitative appraisals that they would be "likely" or "very likely" to visit an SWTD attraction ($n = 26$), the only reasons they gave were entertainment value (96.2%) and/or that it would be especially enjoyable for children (34.6%). They made comments such as that their daughter would love it because she was going to a marine biology camp and that they had done it before in the Bahamas and would do it again. However, 15.4% of this group mentioned that they knew about the related animal welfare concerns. The only respondent to mention *Blackfish* when commenting on the SWTD attraction stated the attraction remained "awesome" despite what he had seen in the documentary.

Among survey respondents "unlikely" or "very unlikely" to visit an SWTD attraction ($n = 51$) and who provided qualitative feedback, the three dominant reasons for their choice were animal welfare concerns (56.9%), lack of entertainment value (21.6%), and human welfare concerns (9.8%). The qualitative feedback included statements that they would only swim with wild dolphins, that "dolphins should be free", that their daughter had experienced a skin infection after her human-dolphin interaction at another SWTD attraction, that they were worried about male dolphins "getting frisky", and that they would rather go to the beach. Of this group, despite their own opposition to the attraction, 11.8% said they would visit with children, as it was more entertaining for that age group. One tourist said she was "unlikely" to visit this type of attraction again, but she "loved it" when she did it before. No respondents mentioned *The Cove* in the unprompted qualitative feedback. A summary of all qualitative responses is detailed in Table 4.

## By accommodation type

Tourists staying in all-inclusive resorts were significantly more "unlikely" to visit a potential SWTD attraction than respondents in other accommodations ($p < 0.001$). Those staying in all-inclusive resorts were also significantly less interested in visiting MMP killer whale

**Table 4  Summary of qualitative opinions offered by TCI tourists on captive cetacean attractions.**

| Visitation likelihood (%) | | MMP | | SWTD | |
|---|---|---|---|---|---|
| | | likely/very likely to visit ($N = 5$) | unlikely/very unlikely to visit ($N = 48$) | likely/very likely to visit ($N = 26$) | unlikely/very unlikely to visit ($N = 51$) |
| **Negative attitudes** | Animal welfare concerns | 20.0 | 72.9 | 15.4 | 56.9 |
| | Not entertaining | – | 10.4 | – | 21.6 |
| | Human welfare concerns | – | 4.2 | 3.8 | 9.8 |
| | Overly commercial experience | – | 14.6 | – | 3.9 |
| | Conservation concerns | – | 4.2 | 3.8 | 2.0 |
| | Attractions too costly | – | – | – | 2.0 |
| | Unclear reasoning | – | 4.2 | – | 5.9 |
| **Positive attitudes** | Entertaining | 100.0 | – | 96.2 | 2.0 |
| | Appropriate for children | 20.0 | – | 34.6 | 11.8 |
| **Influence of media on opinions** | Cited media influence | – | 16.7 | 3.8 | 2.0 |
| | Stated they had seen *Blackfish* | – | 14.6 | 3.8 | 2.0 |

shows ($p < 0.001$). For this variable, and those that follow, a more detailed summary of tourist visitation likelihood is found in Table 5.

## By age
Interest in SWTD attractions decreased with age, with older participants more "unlikely" ($p < 0.001$) to visit. There were no significant differences for MMP killer whale shows on this criterion.

## By country of residence
Significant groupings ($p = 0.001$) were reported for preference toward visiting an SWTD attraction. Respondents from the US were the most positive and were "likely" to visit, compared to Canadians who fell between "likely" and "unlikely", and those from other countries who were generally "unlikely" to visit. After the Bonferroni correction, no significant differences were found between tourists from different countries for visiting the MMP killer whale show.

## By gender, parental status, preference for cruise tourism, and trip frequency to TCI
There were no significant differences in interest in SWTD attractions or MMP killer whale shows by gender, parental status, preference for cruise tourism, or frequency of visitation to the TCI.

## DISCUSSION
### Public opinion of SWTD attractions
A majority of tourists to the TCI supported the introduction of an SWTD attraction. The figure of support (60.3%) was, however, below the 70.2% found in an earlier industry survey (*Alliance of Marine Mammal Parks and Aquariums, 2005*). The reason for support

**Table 5   Visitation likelihoods of TCI tourists to captive cetacean attractions by demographic group.**

| Visitation Likelihood (%) | MMP | | SWTD | |
| --- | --- | --- | --- | --- |
| | likely/very likely to visit | unlikely/very unlikely to visit | likely/very likely to visit | unlikely/very unlikely to visit |
| **Accommodation Type** | | | | |
| All-inclusive | 48.7 | 51.3 | 24.8 | 75.2 |
| Other | 69.0 | 31.0 | 49.4 | 50.6 |
| **Interest in cruise tourism** | | | | |
| Interested | 42.6 | 57.4 | 67.0 | 33.0 |
| Not interested | 36.6 | 63.4 | 56.0 | 44.0 |
| **Residency** | | | | |
| USA | 41.7 | 58.3 | 65.7 | 34.3 |
| Canada | 34.2 | 65.8 | 50.0 | 50.0 |
| Other | 22.2 | 77.8 | 22.2 | 77.8 |
| **Age (yrs.)** | | | | |
| 18–29 | 37.5 | 62.5 | 70.0 | 30.0 |
| 30–39 | 41.4 | 58.6 | 69.0 | 31.0 |
| 40–49 | 50.6 | 49.4 | 68.2 | 31.8 |
| 50–59 | 28.6 | 71.4 | 58.7 | 41.3 |
| 60–69 | 42.1 | 57.9 | 48.7 | 51.3 |
| 70+ | 11.8 | 88.2 | 35.3 | 64.7 |
| **Gender** | | | | |
| Female | 35.3 | 64.7 | 59.4 | 40.6 |
| Male | 44.8 | 55.2 | 62.3 | 37.7 |
| **Visits to TCI** | | | | |
| One | 43.6 | 56.4 | 64.0 | 36.0 |
| Multiple | 33.3 | 66.7 | 56.9 | 43.1 |
| **Parental Status** | | | | |
| Has children | 44.7 | 55.3 | 64.7 | 35.3 |
| Has no children | 34.6 | 65.4 | 56.9 | 43.1 |

in our study was overwhelmingly that such an attraction would be entertaining, with some respondents specifically mentioning their children's potential interest. The earlier, higher figure came from a study that potentially introduced motivated bias by asking respondents to agree or disagree with the statement: "I would be interested in swimming with dolphins in a safe, legal and permitted environment at a marine life park, aquarium or zoo" (*Alliance of Marine Mammal Parks and Aquariums, 2005*). The figure found in our study was also above the percentages found in several previous public opinion studies of tourists, where majorities of respondents reported not favouring visiting captive dolphin attractions (*Draheim et al., 2010*; *Luksenburg & Parsons, 2014*). The demographics of our participants were similar to *Luksenburg & Parsons (2014)*, where 59% of tourists surveyed in Aruba were from the USA. It is possible that, despite Luksenburg & Parsons' best efforts to avoid bias, their use of extensive closed-questioning introduced motivated, ingratiation, and/or social desirability bias.

Alternatively, the differences between the findings from these studies and ours may be due to the sampling locations and approaches. *Draheim et al. (2010)*, *Luksenburg & Parsons (2014)*, and ourselves all surveyed tourists, but in different resorts. Visitors to different locales can have different attitudes, as is the case for tourist perceptions of the natural environment, which differ by island in the Caribbean region (*Uyarra et al., 2005*). Our study is also less comparable to *Miller et al. (2013)*, where visitors to a captive marine mammal attraction were surveyed while in attendance or *Patterson (2010)*, where the surveys were conducted with volunteers for a high-end whale watching and eco-tourism outfit. These studies sampled sections of the public predisposed to visiting cetacean attractions, while our study sampled a section of the general public for whom no predisposition towards or against visiting captive cetacean attractions can be assumed.

While previous research has cited education, conservation, and welfare benefits as the reasons the public gave for visiting captive cetacean attractions, in this study, the only motivation mentioned by respondents for visiting SWTD attractions was their entertainment value. This suggests that the polls that found that at least 80% of respondents see educational and conservation values in captive cetacean attractions have over-asserted these attitudes (e.g., *Alliance of Marine Mammal Parks and Aquariums, 2005*; *Miller et al., 2013*). This difference may be due to potential ingratiation bias in the survey by *Miller et al. (2013)*, where statements like "this experience was educational" were put to respondents while they were visiting the attraction. It would be uncomfortable for a respondent to respond negatively to this statement while talking to a surveyor they might suspect has a working relationship with the attraction. *Jiang, Lück & Parsons (2007)* similarly found that conservation value was not greatly attached to captive cetacean attractions by visitors, but even their paper, openly sceptical of the educational value of such attractions, still found that visitors offered education as a reason for their attendance. *Jiang, Lück & Parsons (2007)* also specifically asked about education opportunities, which might have introduced motivated bias through the survey questions, as a result of leading respondents to assign more weight to an issue than they might have independently. We expect that researchers seemingly sceptical of a perceived value to the public would not intentionally insert bias that caused that value to be highly reported by respondents, illustrating how hard it can be to design written, closed questions that do not influence the participant. Additionally, the benefits to human health claimed in some research may not be a valid reason for maintaining and developing SWTD attractions, as respondents in our research only identified associated threats to human well-being from visiting such attractions.

Furthermore, 39.7% of tourists surveyed here were not in favour of visiting SWTD attractions, primarily citing dolphin welfare concerns, where reasoning was provided. These viewpoints cast doubt on the interpretations of the *Alliance of Marine Mammal Parks and Aquariums (2005)* survey, where AMMPA stated that the general public believed animal welfare was high at such attractions. In that survey, 95% agreed that "the people who care for the animals at marine life parks, aquariums and zoos are committed to the welfare of the animals" (*Alliance of Marine Mammal Parks and Aquariums, 2005*), a question more focused on the capability of the trainers than the condition of captive animals. The wording likely introduced the motivated bias of the researchers. Our results are closer to those found

by *Jiang, Lück & Parsons (2007)* in Canada, where a major reason for non-visitation was animal welfare concerns.

In our study, overall opposition to an SWTD attraction was noted for those staying in all-inclusive accommodation, tourists residing outside of the US and Canada, and older adults. The lack of appropriate qualitative data offered by most respondents makes it hard to fully explain their opposition. Whatever their reasoning, the opinions of these demographic groups have implications wherever they are present. All-inclusive tourism models are particularly popular in the Caribbean (*Brida & Zapata-Aguirre, 2010*), with just above 50% of tourists not from the US (Caribbean Tourism Organization, 2014), and the average age of visitors from the US being over 40 (*International Trade Administration, 2014*). All-inclusive resorts provide entertainment for their guests, and for countries like the TCI, where these resorts are among the biggest individual employers (*Allen, 2013*), there are limits to the market for SWTD attractions.

In general, TCI tourists were willing to visit an SWTD attraction, even when aware of the associated animal welfare concerns amplified in recent media. *Jiang, Lück & Parsons (2007)* also found this to be the case among the Canadian public. Researchers and advocates opposing dolphin captivity may still see an opportunity to influence public opinion, however, in the relatively low level of human welfare and dolphin conservation concerns recorded in this study. *Draheim et al. (2010)* noted that tourists in the Dominican Republic were similarly unaware of welfare and safety concerns, with 75% of their sample not seeing swimming with dolphins as dangerous. Their study also found that, when required to provide a closed answer, over 80% of tourists placed weight on dolphin conservation issues. Without prompting from close-ended survey questions, dolphin conservation was barely identified as an issue by TCI tourists. If this was due to a lack of awareness rather than apathy, then there is potential to increase public knowledge of both conservation and welfare issues.

## Low public opinion of MMP killer whale shows

TCI tourists' overall attitude toward MMP killer whale shows was largely negative. The 60.9% who identified as not likely to visit such attractions roughly correlates to a recent survey where closed-ended questions found 50% of a sample of the general public opposed killer whale captivity (*Edge Research, 2014*). It is possible that, because of their use of telephone interviews conducted by professional surveyors, the surveyors managed to reduce some ingratiation and social desirability bias (*Rossiter, 2009*). There is also the possibility that motivated bias introduced through question design had a lesser impact, as public opinion of killer whale captivity in MMPs was already strongly formed, possibly due to exposure to more media on the welfare of captive killer whales than captive dolphins.

Tourists not staying in all-inclusive resorts were the only respondent demographic to clearly identify as positive towards visiting an MMP killer whale show, but there were not enough qualitative responses to explain why. Of the reasons given, the strongest concern was for animal welfare. Education, conservation, and human welfare benefits were not cited as reasons for wanting to visit. Conservation concerns were not mentioned as a deterrent. Again, this contrasts with previous surveys that may have introduced pro- or anti-captivity

researcher-motivated bias through their use of close-ended questions, such as those that have found a wide range of respondent agreement (56–97%) that these attractions offer educational experiences (e.g., *Alliance of Marine Mammal Parks and Aquariums, 2005*; *Jiang, Lück & Parsons, 2007*; *Edge Research, 2014*).

Media influence had noticeably more impact on respondents' opinions of MMP killer whale shows than SWTD attractions, with several citing having watched *Blackfish* as their reason for not wanting to visit such an attraction. This influence is supported in the results of a recent survey, which showed that 73% of the US public learned about killer whales via the media (*Edge Research, 2014*). It is also reflected in the dramatic fall in the stock market value of North America's primary provider of killer whale attractions, which has been blamed on negative publicity and resultant decreasing visitor numbers (*Peterson, 2014*; *Huggan, 2017*). In 2016, the same provider announced the end of their captive breeding programme and therefore the eventual end of captive killer whale shows at their attractions (*Hacket, 2016*). However, while the documentary extensively covered the human welfare issues associated with training killer whales, TCI tourists' qualitative responses rarely explicitly identified these issues, even when mentioning *Blackfish*.

## Shifting public opinion of SWTD attractions and MMP killer whale shows

The issues contributing to the public opinion of dolphin and killer whale captivity are similar, yet the respondents here were more likely to visit SWTD attractions than MMP killer whale shows. In the qualitative responses, media influence was cited less frequently for SWTD attractions as a factor in potential visitation. The showing of *Blackfish* on well-watched television outlets is credited for broadening the media profile of the negative issues associated with MMP killer whale shows, especially given the deaths of trainers highlighted in the film (*Huggan, 2017*; *Parsons & Rose, 2018*).

A similar shift in public opinion could be expected if a member of the public were seriously harmed at an SWTD attraction (*Hunt et al., 2008*; *Rose, Parsons & Farinato, 2009*). Indeed, shifts in public opinion have already been credited for the closures of the last United Kingdom captive dolphin attractions in the 1990s (*Hughes, 2001*) and a facility in the Bahamas in 2014 (*Lowe, 2014*). Pushback against a plan to construct an SWTD attraction in Arizona, USA led to a petition with over 170,000 signatures (*Milman, 2016*; *Dee, 2016*). In these cases, dolphin welfare has primarily driven public opinion, though recent opposition has cited bites from dolphins and "incidents that resemble sexual assault" (*Milman, 2016*). Nevertheless, the captive cetacean industry continues to invest in infrastructure and propose new attractions. Approximately 25 additional SWTD attractions have been proposed for the Caribbean region (*Rose, Parsons & Farinato, 2009*), including the two in the TCI. Policy-makers, governments, and tourist attraction developers need to be aware of potential negative shifts in public opinion of SWTD attractions, as they would likely cause the same drop in visitation as for MMPs.

Our results could also be interpreted as the public believing that dolphin welfare in SWTD attractions is higher than for killer whale welfare in MMPs. Yet this interpretation is actually contrasted by our qualitative data, which suggest that dolphin welfare is not at

the forefront of public consciousness. Very few of our participants commented on captive cetacean welfare and those that did mentioned killer whale captivity rather than that for dolphins. Either the public care less about the welfare of captive dolphins than of killer whales, perceive dolphins to be better-suited to captivity, leading to better welfare, or have not been as exposed to the welfare issues surrounding dolphin captivity. Our study lacks the data necessary to determine which scenario is most likely and further research is necessary to determine the root of the lack of public consciousness over dolphin captivity.

## Alternatives to captive cetacean attractions

Aquariums and botanical gardens, rated by TCI tourists as significantly more desirable than MMP killer whale shows and similarly desirable to SWTD attractions, have been shown to provide educational and conservation value (*Parsons & Muhs, 1994*; *Falk & Adelman, 2003*; *He & Chen, 2012*), with less debate than for captive cetacean attractions. Where possible, wild whale and dolphin-watching tours may be better attractions to endorse as they have, in some studies, been found to have fewer negative conservation and animal and human welfare issues (*Jiang, Lück & Parsons, 2007*). Research in Aruba (*Luksenburg & Parsons, 2014*), the Dominican Republic (*Draheim et al., 2010*), and Belize (*Patterson, 2010*) has shown that visitors would prefer wild cetacean encounters to captive ones.

## Advantages and limitations of the photo elicitation methodology

The relatively low qualitative response rate from participants impeded our ability to explain the respondents' reasoning behind their quantitative Likert ratings. All of the other public opinion studies of captive cetacean attractions reviewed here did draw specific conclusions about whether entertainment, educational, or conservation value, or human or animal wellbeing, were reasons for visitation or non-visitation, because they received a complete response rate to their worded, closed questions. Yet we know these worded questions can also be limited, likely more so, due to the opportunities for introducing bias that distorts responses. In some cases, it might be easier to write more neutral closed survey questions, but with emotionally-sensitive topics like animal captivity and welfare, it is difficult. The benefit of photo elicitation is that it leaves the respondent open to responding how they want to, decreasing potential bias.

The results of this study, for instance, do not reveal whether TCI tourists believed an SWTD attraction would be educational or improve their wellbeing, or otherwise, but this ambiguity can be valuable. The lack of definitive qualitative findings from respondents on the potential educational, conservation, and human wellbeing benefits of SWTD attractions, or of animal and human welfare issues, is because the respondents guided this research. Many of the issues previously highlighted by researchers were simply not at the forefront of TCI tourists' minds when considering which attractions they would like to visit. Conclusions drawn elsewhere, therefore, may have been influenced by researcher-introduced bias, preventing a true snapshot of public opinion. While the lack of qualitative responses to photo elicitation limits the explanatory power of our quantitative findings, the responses that were elicited, especially on the entertainment value of SWTD attractions, do begin to explain our data. Follow-up research, attempts at replication, and comparative

case studies should look to elicit more extensive open-ended responses from participants. These could use an approach that compels a comment for each photograph, as long as care is still taken to avoid introducing researcher bias.

One success of this methodology was not initially revealing the full research aims to respondents, reducing bias. The substantial differences between the opinions of TCI tourists on cetacean captivity and those found in several similar surveys is likely down to our accounting for the biases listed in Table 1. Many of the other surveys did not describe attempts to reduce these biases. Yet our selection of photographs may have remained an issue. The photographs used in this approach, though carefully selected with a theoretically-grounded approach, could have inherently influenced respondents. The image used for the SWTD attraction (Fig. S1), for instance, is a close-up of a child smiling while swimming with a dolphin, while the image for the MMP killer whale show (Fig. S1) is a more distant photograph with the faces of spectators out of focus. As photographs with smiling subjects tend to indicate positive experiences (*Miles & Johnston, 2007*), the SWTD attraction image is more likely to have attracted additional positive responses, irrespective of general opinions of the attraction, and perhaps has artificially inflated our figure for respondents' likelihood to visit SWTD attractions. No respondents commented on the image content (Fig. S1), however, suggesting this was likely not a major issue here. Still, bias introduction through photograph selection cannot be ruled out. Further photo elicitation studies on the influence of photographs of subjects with varying expressions, or photographs where facial expressions were not shown, would give further context to the value of the quantitative results presented here. To more definitively determine the strengths and limitations of the photo elicitation methodology, a follow-up study comparing photo elicitation in parallel to traditional survey methods would be useful.

## CONCLUSIONS

There is no consensus on public opinion of captive cetacean attractions. Underestimation of unintentional researcher bias in study design and probable attempts to deliberately guide respondents toward the outlooks of those conducting or commissioning research have led to a spectrum of reported opinion. For this study, we selected a photo elicitation approach and attempted to account for all forms of bias. Our findings suggest that previous claims of public support for MMP killer whale shows have likely been overstated, as have assertions of both opposition to and support for SWTD attractions. While the photo elicitation approach employed here has its own limitations, the method avoids the insertion of the most severe researcher-driven biases and hands the initiative for evidence production on public opinion of captive cetacean attractions to the public themselves. Policy-makers and developers should not base their decisions on licensing and building captive cetacean attractions on the outcomes of public opinion studies without scrutinising the validity of how public opinion was surveyed.

Researcher-introduced bias seems to have been a particular issue in over-assigning some values of captive cetacean attractions to the public. The lack of respondent comments on the educational or conservation value of captive cetacean attractions suggests that previous

studies have erroneously introduced these as major issues of public focus through inserting survey questions on these issues. With some of these values disputed by researchers, there is a case for giving less weight to them as factors in decision-making on the development of captive cetacean attractions. Their entertainment value, which here was already diminished for MMP killer whale shows, could diminish for SWTD attractions if the public becomes concerned with the conservation, animal welfare, and human welfare issues associated with such facilities. There would seem little long-term public value to captive cetacean facilities and their further development should perhaps be reconsidered.

Ultimately, all involved in proposing or opposing cetacean captivity require a better baseline of public opinion toward MMP killer whales shows and SWTD attractions. Future research must involve a greater effort to address methodological biases. This can be achieved through mixed-methods approaches that still allow researchers to quantitatively assess the elements of public opinion they are interested in, but which first permit respondents to provide qualitative feedback using their own voice. The photo elicitation approach used here was partially successful in doing this, but was limited by the number of qualitative responses fostered. Best practice might be to follow a similar approach, but ask additional, neutral open-ended questions at the start of the survey, or to compliment it with other qualitative approaches (e.g., interviewing) that allow a more in depth investigation of quantitative findings.

## ACKNOWLEDGEMENTS

We gratefully acknowledge the key logistical support provided by The School for Field Studies (SFS) Center for Marine Resource Studies. We also thank Dr. Chris Parsons, Dr. Naomi Rose, and Peter Craig for their extremely helpful comments on previous drafts of this transcript, as well as to a number of anonymous reviewers. We thank Morag Taite for her ArcGIS expertise in constructing Fig. 1. Finally, we are indebted to the students of The SFS Center for Marine Resources Studies Spring '14 Program who did an excellent job conducting the surveys. This research project would not have been possible without their participation, diligence, professionalism, and dedication.

### Funding
This work was supported by the School for Field Studies Center for Marine Resource Studies and the Turks and Caicos Department of Environment and Marine Affairs. The funders had no role in study design, data collection and analysis, decision to publish, or preparation of the manuscript.

### Grant Disclosures
The following grant information was disclosed by the authors:
School for Field Studies Center for Marine Resource Studies.
Turks and Caicos Department of Environment and Marine Affairs.

## Competing Interests

The authors declare there are no competing interests.

## Author Contributions

- Sophia N. Wassermann and Edward J. Hind-Ozan conceived and designed the experiments, performed the experiments, analyzed the data, prepared figures and/or tables, authored or reviewed drafts of the paper, approved the final draft.
- Julia Seaman analyzed the data, prepared figures and/or tables.

## Human Ethics

The following information was supplied relating to ethical approvals (i.e., approving body and any reference numbers):

The supporting institution, the School for Field Studies, did not have an Institutional Review Board at the time, nor did any institutions in the Turks and Caicos Islands (TCI) provide IRBs. The study followed all legal and ethical guidelines for conducting research in the TCI.

## Data Availability

Raw data is provided in the Supplemental Materials.

## Supplemental Information

Supplemental information for this article can be found online at http://dx.doi.org/10.7717/peerj.5953#supplemental-information.

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
