# Peer review of "Reassessing public opinion of captive cetacean attractions with a photo elicitation survey"

_PeerJ, doi:10.7717/peerj.5953_

## Round 0.1 · original submission · Major Revisions

Both reviewers found your manuscript to be interesting and relevant. However, they have made a number of constructive criticisms with which I concur. I would therefore encourage you to carefully address the issues raised, since your manuscript would be substantially improved as a result.

·

Basic reporting

This is an interesting study that adds to the literature on public attitudes towards cetaceans in captivity. The literature on captive marine mammals has been well researched.

I think, however, that assuming that using images leads to less bias may be wrong. As the researchers note, positive feelings can be generated by images with smiling faces. However, these sections of the manuscript can easily be toned down to be more cautious.

With this in mind, the researchers should pay particular attention to the abstract (line 46-55) - the discussion is summarized with a certainty that is certainly not warranted.

Some recent studies (e.g. Whitney Naylor et al.) have shown that public attitudes shift according to what type of captive display setting animals are in e.g. there is more sympathy for rehabilitation and research facilities than dolphin show facilities. This study is possibly picking up some of that difference in attitudes (i.e., depending on facility type).

It would have been interesting to find out where the tourists thought the animals in swim with facilities came from. For example, some tourists I questioned thought that dolphins in swim-with facilities were free to go out to sea at night, and came in to interact with tourists during the day because they wanted to... thus, they thought they were actually "wild" dolphins.

Anyway, this is an interesting study, and I suggest a few minor edits just to be more cautious.

One minor typo/mistake:
Line 77 - Parsons, E. C. M., & Muhs, K. (1994). Note that this was written by Chris (Christine) Parsons the science communicator not Chris (ECM) Parsons the marine mammalogist

Experimental design

I would be somewhat more cautious when discussing the idea that surveys using images result in less bias.

As noted above, the latter lines of the discussion should be re-written (~45-55). There is an untested assumption that using images leads to less bias than check box type questionnaires. This simply cannot be stated with certainty.

For example, Megan Draheim et al. found in several studies that the type of image of a coyote viewed by those questioned resulted in different levels of conservation concern e.g. images of coyotes playing led members of the public to produce substantially different results to images of coyotes howling. Images can substantially influence attitudes - that's why they are used in marketing campaigns all the time. Some citation of such studies would be useful in this paper.

What would have been really helpful to assess the methodology would have been a control sample with no images being used, to see if there was a significant difference in response with or without images (perhaps this could be added to the paper as possible future studies?).

As a result, I think that statements like line 530-535 are very valid - smiling faces and happy people are probably more likely to lead to a positive response. Suggesting a follow up study without images or a dual image/without image study would be a good idea.

Karaffa et al. found that rebranding species names to sound cute or patriotic (e.g. changing coyote to native american song dog) lead to a substantive leap in conservation concern. So even little tweaks like this can effect public interest/attitudes.

In short, there is no evidence that using images results in bias free surveys. In fact, there are some studies that suggest that images could potentially result in bias.

Line ~380. Personally, I think that a lot of the differences could be explained by different demographics and tourist types. For example, Katheryn Patterson's tourists in Belize were relatively high end, environmentally-aware ecotourists. If the majority of tourists in Turks and Caicos were cruise ship mass tourists, that might explain some of the differences. The demographics of tourists in the Dominican Republic are probably closer to those in the Turks and Caicos (e.g. mostly inclusive resort beach tourists in Draheim et al. ).

Several factors influence attitudes to captive marine mammals, including education, political affiliation, affluence, whale-watching experience, NGO membership etc. Differences in any of these factors might effect differences between populations.This section (Line 380~) could perhaps be expanded to discuss all the various factors that have been found to influence attitudes towards captivity.

In the US these factors can be seen regionally, for example, in the various marine theme parks: SeaWorld in San Diego and Florida and losing tourists, but not so much for SeaWorld San Antonio.

With a sample of 259, the 95% margin of error is +/- 6.1% approximately. This should be included in the methodology.

As noted above, the discussion section of the abstract (lines 45-55) needs rewriting. At present emphatic statements are made which cannot really be supported. It would be better for the authors to moderate their statements. EG just because their study does not match the results of other studies, does not mean they these other studies are "biased" or "misleading", as noted above, demographic differences and sample error could explain a lot of the variability. Just describing their results and noting that that they are slightly different to some other studies, which might be because of variety for factors including methodology and demographics etc, would be a more temperate and less controversial conclusion.

Validity of the findings

As noted above, the findings of the study are perfectly valid, but some of the interpretation and conclusions need to be edited slightly to perhaps be somewhat more cautious.

Additional comments

I enjoyed reading this paper. It was very well written and well researched, and would provide an interesting addition to the literature on public attitudes to cetaceans. I have just a few minor suggestions for the researchers to be a bit more cautious in their interpretation.

The researchers should be aware that when published this paper might get a lot of press attention as it's quite a hot topic issue. They should be wary that the press and activists from either side of the captivity issue might take sections out of context, so they should try to "bad press proof" the paper as much as they can.

·

Basic reporting

In general, the use of English language, grammar and sentence structure was very good in this paper. The referencing itself was well organised and extensive, and the raw data was provided (although analyses would be very hard to replicate as there are many values without explained units or how they were calculated).

The style of the paper needs to be significantly improved. I found many examples of poor scientific writing style, where for example emotive or imprecise phrases were used (e.g. “…identified as not keen to visit such attractions roughly correlates to…”, Line 444). This was also linked to the tone behind the writing, which I felt had an agenda to discussing the literature and results in light of opposing captive cetacean practices (e.g. “Those promoting the education, conservation and welfare benefits of SWTD… should… take note”, Line 391).
It was my understanding that although the authors self-identified as “anti-captivity”, they wanted to avoid the bias seen in previous studies and conduct a balanced study on the topic. I have made suggestions using the comment boxes on the PDF where I believe that counter-explanations for results should also be provided. It may sound like I am on the pro-captivity side (I assure you I’m not, I work on both “sides” because I focus on the animal’s welfare), but I have played Devil’s Advocate and made these suggestions in order to help the authors produce a clear, well-balanced paper where the results speak for themselves. As it is, I feel that the study’s interesting results get a bit lost among the frequent reference to the controversial debate and purported claims.

The authors clearly explained the background and associated issues of survey studies and methodologies, which will be appreciated by the less knowledgeable readers. I felt there was literature lacking regarding the measurement of dolphin welfare in captivity, for which there have been a lot of studies published over the last 5 years. Farm animal welfare techniques allow us to understand what animals are feeling, and focus on “animal-based measures”, i.e. the behaviour, health etc of the animal. Although not the main point of this study, such studies on dolphins should be listed to give the readers an idea about what dolphin welfare is and how it can be recognised.

To summarise the findings on dolphin welfare so far: their cognitive bias has been measured, showing that some were in good and some in poor welfare, and that it was related to their social bonds. They have been found positively anticipate non-food trainer interactions, and that social play is a good welfare indicator. There has also been a full assessment proposed which can objectively measure dolphin welfare. In light of the authors’ comments on “research by those working at captive cetacean attractions”, although these publications do indeed usually include an investigator from a dolphin facility, this is expected as the research needs long-term access to the dolphins. There are nearly always collaborating non-industry scientists and the research is peer-reviewed so can be considered objective.

Brando, S., Broom, D. M., Acasuso-Rivero, C., & Clark, F. (2017). Optimal marine mammal welfare under human care: Current efforts and future directions. Behavioural processes.

Clegg, I. L. K., Rödel, H. G., Boivin, X. & Delfour, F. (2018). Looking forward to interacting with familiar humans: dolphins’ anticipatory behaviour indicates their motivation to participate in specific events. Applied Animal Behaviour Science. 22, 85-93

Clegg, I. L. K. & Delfour, F. (2018). Cognitive judgement bias is associated with frequency of anticipatory behaviour in bottlenose dolphins. Zoo Biology, 37, 67-73

Clegg, I. L. K., van Elk, C. E., & Delfour, F. (2017). Applying welfare science to bottlenose dolphins (Tursiops truncatus). Animal Welfare, 26, 165-176.

Clegg, I. L. K., Rödel, H. G. & Delfour, F. (2017). Bottlenose dolphins engaging in more social affiliative behaviour judge ambiguous cues more optimistically. Behavioural Brain Research, 322, 115-122.

Clegg, I. L. K., Borger-Turner, J. L., & Eskelinen, H. C. (2015). C-Well: The development of a welfare assessment index for captive bottlenose dolphins (Tursiops truncatus). Animal Welfare, 24(3), 267-282.

Makecha, R. N., & Highfill, L. E. (2018). Environmental Enrichment, Marine Mammals, and Animal Welfare: A Brief Review. Aquatic Mammals, 44(2), 221-230.

Serres, A., & Delfour, F. (2017). Environmental changes and anthropogenic factors modulate social play in captive bottlenose dolphins (Tursiops truncatus). Zoo biology, 36(2), 99-111.

Experimental design

The research described in this paper is well-designed and executed, and certainly within the scope of the Journal. It addresses the need to understand tourist opinion on the subject in this area of the Caribbean, given the recent law changes and proposed developments. It also has wider implications for understanding general public feeling to captive dolphins and killer whales, although I would suggest that the authors acknowledge more clearly the limitations regarding how representative their results are of the general population. The methods were described clearly and thoroughly, with only a few comments where I thought they could be clarified or rephrased. The surveys were carried out ethically and a good sample size of respondents was obtained.

I also do not understand why there were two figures made available to me showing the 6 photo grid: one was Fig.S1, which was mentioned in the text, and then the other was a similar grid with slightly different photos and the child’s face blacked out, but it was not mentioned in the text. This file was labelled: “cc_photo_representatives”. Could you clarify its purpose?

Validity of the findings

The dataset itself is extensive and was treated appropriately in preparation for analyses. Suitable statistical tests were used for the analysis and were in line with other similar social science work. I commented on the p-value reporting, where I don't see the need to report the different values below 0.001 (simply put p<0.001), and there was one instance where the p-value seemed to have been used to discuss effect size, which should be corrected.

In my opinion, the discussion section has a good foundation but much work is needed to improve the style and make the discussions more well-rounded. Alternative explanations for the findings should always be discussed, and I felt that again the tone was leading towards those with “anti-captivity” opinions. The majority of the readers of this paper won't have strong pro or anti captivity opinions, and the repeated statements bringing up the debate won’t lend to their engagement and appreciation of the paper. At the moment, the flow is disrupted and your valuable findings are being lost in translation because they are constantly being framed within this debate. If you write objectively and scientifically, where all inferences are backed up by peer-reviewed research, your results will have a much larger impact.

A few areas of the discussion would really benefit from expansion, and I have commented specifically on these in the PDF. Again, I think it’s important to mention the progress that is being made into measuring cetaceans welfare, where last year a study found that some dolphins in a group were objectively in good welfare in captivity, as measured through their cognitive bias. Given that your respondents often did not support SWTD on welfare grounds, how might new welfare research impact this? I realise this is not the main discussion of the paper, but I think the discussion will really benefit from including studies from adjacent disciplines, as this is what is likely to be informing the tourists’ responses.

Further discussion on the reasons for the different findings for killer whales and dolphins would also be valuable, as I think this is one of the most interesting results. Smaller dolphins (e.g. Tursiops) are thought to be more adaptable and fare better in captivity than killer whales, mainly due to their less aggressive tendencies, their fission-fusion groups (as opposed to rigid family structures), and their smaller size.

As mentioned earlier, the limitations of the research should be better outlined, and speculation over the findings should be written as such and positioned appropriately- for example, the current last sentence of the discussion negates the study’s approach, should be rephrased more cautiously and put earlier on when evaluating the general methodology.

Additional comments

This study looking into the opinion of TCI tourists to captive cetacean facilities fills a current gap and will contribute to our general understanding of the public’s opinion into this practice. The methods were appropriate and well-thought out, well executed and the data was handled correctly. The use of English and grammar was very good, with some comments on parts that need clarification, but more importantly the authors need to make major improvements to write in a more scientific style throughout the paper. They should focus on writing a well-rounded, balanced study of this topic, where arguments and counter-arguments are given for each point, where appropriate of course. In this way, the flow of writing will be improved and the paper’s results will reach a wider audience and be more readily accepted. I hope the authors decide to make the revisions and resubmit.

---

## Round 0.2 · Minor Revisions

Thank you for addressing most of the reviewers' previous comments. However, both reviewers have raised some issues that still need to be addressed.

·

Basic reporting

A couple of minor suggestions for the introduction:

For the introduction Miller et al., 2013 is used as evidence that dolphin shows and interaction programs gave educational and conservation benefits. However, there was no statistically significant difference between those tourists who had actually viewed or interacted with dolphins and the (control) group of park visitors that had not, in terms of knowledge, attitudes towards conservation or conservation intentions - therefore being able to see captive cetaceans actually had no observable effect on education or conservation-oriented behavior. It is more likely that changes in awareness were due to signage and other non-animal materials that tourists were exposed to at the park, rather than the animals/animals shows per se.

Jiang et al., 2007 is used as a citation (in the introduction & discussion) that marine parks are educational, although the authors say that they tested whether tourists were "aware of educational opportunities" or "whether they felt" that they had been education, but did not test whether knowledge had increased. It's a subtle difference, but an important one.

Wild captures for marine parks, especially in China and Russia are rather larger than outlined in the introduction, e.g. this hot off the press story: https://www.reuters.com/article/us-china-marineparks-insight/tidal-wave-of-chinese-marine-parks-fuels-murky-cetacean-trade-idUSKCN1M00OC
Perhaps quantify the huge number of facilities in China that are opening, and being stocked by wild captures?

Denham & Parsons (in Tourism in Marine Environments, in press) has a recent survey of attitudes to cetaceans in captivity, including by types of facilities (primarily US & India participants - if you contact Whitney or Chris they could probably supply you with a copy of the paper).

The added Clegg et al. studies are very controversial - they have been cited as evidence that dolphins "are happy" and show signs of great welfare by have been heavily criticized because of over-interpreting results. For example, the studies to show anticipatory behavior towards trainers - as is seen in many domesticated animals - but there is a big gap between this and saying that they have good welfare or "are happy" as was stated in the press release for this paper.

One good measure of welfare is mortality rates, it might be worth mentioning the Jett and Ventre found that captive orca mortality rates were poor compared to wild populations https://onlinelibrary.wiley.com/doi/abs/10.1111/mms.12225 . Might be worth inserting this study as it has thoroughly reviewed science.

Comparisons of bottlenose dolphin mortality to wild populations are rather controversial at the moment as the best-studied wild population for mortality rates is not doing well (because of pollution, oil spill impacts etc etc), so saying that captive animals do as well as Sarasota dolphins (or starving, so to go extinct southern resident killer whales) is faint praise indeed. It's like comparing human health against a control population from Haiti...

Line 826: "Big Changes at Sea Sworld. "

Experimental design

No further revisions suggested.

Validity of the findings

No further revisions suggested.

Additional comments

No further revisions suggested. Really interesting paper.

·

Basic reporting

This has been improved with the author's revisions, thank you.

Experimental design

Methods have been made clearer.

Validity of the findings

Conclusions have been toned down, and the findings better aligned with the literature, thank you.

Additional comments

Thank you for your revisions and your responses. I find the manuscript a lot clearer, and the discussion much better.

I still have some comments- added directly onto the manuscript- on the use of language and also regarding a few statements where I still would suggest toning down to make the points more objective, and wouldn't be happy recommending publication until they are altered.

You have done a great job so far with the revisions so I am confident you can amend these last few things!

---

## Round 0.3 · accepted · Accept

Both reviewers have recommended Acceptance.

# ·

Basic reporting

Good

Experimental design

Good

Validity of the findings

Good

·

Basic reporting

All fine- see previous reviews.

Experimental design

All fine- see previous reviews.

Validity of the findings

All fine- see previous reviews.

Additional comments

I agree with all the last changes you made, and applaud you for your diligence in improving this paper!